# Males' perceptions and practices towards maternity care in rural southeast Nigeria: Policy implication of participatory action research for safe motherhood

Irene Ifeyinwa Eze[1,2,3]*, Edmund Ndudi Ossai[1,2], Ifeyinwa Chizoba Akamike[1,3], Ijeoma Nkem Okedo-Alex[1,3], Lawrence Ulu Ogbonnaya[1,2,3], Chigozie Jesse Uneke[2,3]

1 Department of Community Medicine, Alex Ekwueme Federal University Teaching Hospital Abakaliki, Abakaliki, Nigeria, 2 Department of Community Medicine, College of Health Sciences, Ebonyi State University, Abakaliki, Nigeria, 3 African Institute for Health Policy and Health System, Abakaliki, Ebonyi State, Nigeria

* jorenebiz@yahoo.com

## Abstract

### Introduction

High maternal death is attributable to developing countries' health systems and sociocultural factors This study assessed the effect of participatory-action research on males' perception and practice towards maternity care and safe motherhood in rural southeast Nigeria.

### Methods

A pre-post-intervention study design was employed to study 396 male partners of pregnant women selected through cluster sampling in rural communities in southeast Nigeria. Males' perceptions and practices towards maternity care and safe motherhood were assessed using an interviewer-administered five-point Likert scale questionnaire. A community-participatory intervention was implemented comprising advocacy, and training of community volunteers, who then educated male partners of pregnant women on safe motherhood and facilitated emergency saving and transport schemes. A post-intervention assessment was conducted six months later, using the same questionnaire. Good perception and good practices were determined by mean scores >3.0. Continuous variables were summarised using mean and standard deviation, and categorical variables using frequencies and proportions. A comparison of the mean scores pre- and post-intervention mean scores were compared, and the mean difference was determined using paired T-test. Statistical significance was set at a p-value <0.05.

### Results

The perception that male partners should accompany pregnant women for antenatal care had the least mean score at the pre-intervention stage, 1.92 (0.83). However, the mean score increased for most variables after the intervention (p<0.05). The mean score for

**Funding:** The authors received no specific funding for this work.

**Competing interests:** The authors have declared that no competing interests exist.

maternity care practices increased post-intervention for accompanying pregnant women to antenatal care, facility delivery, and helping with household chores (p<0.001), with a composite mean difference of 0.36 (p<0.001). Birth preparedness/complication readiness practices–saving money, identifying transport, skilled providers, health facilities, blood donors and preparing birth kits, were good, with a composite mean score that increased from 3.68 (0.99) at pre-intervention to 4.47(0.82) at post-intervention (p<0.001).

## Conclusions

Males' perceptions and practices towards safe motherhood improved after the intervention. This highlights that a community-participatory strategy can enhance males' involvement in maternal health and should be explored. Male partners accompanying pregnant women to clinics should be advocated for inclusion in maternal health policy. Government should integrate community health influencers/promoters into the healthcare systems to help in the provision of health services.

## Introduction

Maternal survival is a major concern in most developing countries due to the high maternal mortality ratio (MMR). Nigeria contributes only 2.4% to the world's population but in 2008, bore 14% of the global burden of maternal mortality, with MMR estimated at 545 deaths per 100,000 live births [1]. The World Health Organization (WHO) explains the problem of maternal death using a delay model that influences maternal survival and is linked to the health system, socioeconomic and human factors [2]. Such human factors include fatalistic beliefs and attitudes which govern social behaviour, including the central role of men in decision-making [3, 4]. Males, by tradition, are invested with power and wield authority even in the domain of women's health [5–7]. In Nigeria, a lower proportion of employed women compared to men receive payment for their work and have control over their cash earnings and household decisions [8]. Studies conducted in sub-Saharan Africa have revealed low involvement of men in maternal health [9, 10]. Poor males' involvement in maternity care was also noted in earlier studies where less than half of the men studied could mention three or more pregnancy-related danger signs, prepare for birth, accompany their spouses to antenatal care (ANC) and are present during childbirth [11, 12]. A man is involved in maternal health if he is "present, accessible, available, understanding, willing to learn about the pregnancy process and eager to provide emotional, physical and financial support to the woman carrying the child" [13].

Propagating male involvement in maternal health will be in tune with the international conference on population and development (ICPD) call to give importance to males' shared responsibility and to promote their active involvement in responsible parenthood. Male involvement in maternal health is a marker of "gender equity" It is part of social determinants of health which posits the adoption of more equitable gender roles such as joint decision-making, shared household chores and parenting among couples [14]. It has the potential to position man as an agent of positive change. In addition, males' involvement in maternal health is positively associated with access to quality health care [12]. Male partners accompanying pregnant women to access healthcare were positively associated with women receiving care from the medically trained provider for ANC, birth at a health facility, postnatal care (PNC), and seeking care for obstetric complications [12].

Most programmes promoting women's reproductive health have focused mainly on women who traditionally are not decision-makers and whose reproductive choices or outcomes depend on their male partners' financial power and support. In contrast, men who typically serve as gatekeepers of women's reproductive health are left out [9, 13]. Furthermore, insufficient awareness about safe motherhood has been reported in several studies conducted in developing countries, including Nigeria [4, 15], notwithstanding that health information is commonly provided to people, especially at health facilities. These raise questions about whether facility-based dissemination of messages can continue to be the major reliable means of conveying this information. It could be done in new ways that go beyond the health facility by making community and family members, including male partners participate as they play vital roles in healthcare decisions. Also, men's participation in maternal health is an area seldom focused upon by research studies, probably because of the assumption of women's primacy in fertility [9, 16, 17], notwithstanding that decision-making for accessing care is dominated by men [6]. Increased male participation could yield health benefits for men, women and family at large by ensuring the use of ANC, healthy practices during pregnancy, and facility delivery [9]. These highlight the importance of using strategies that involve men as partners and accommodate both genders.

Many programmes have been instituted to address the problem of maternal deaths including the safe motherhood initiative (SMI) launched in 1987 by UN agencies—UNFPA, the World Bank, and WHO in Nairobi, Kenya as a global campaign for the reduction of maternal mortality and morbidity. The key messages of SMI were that every pregnancy faces risk and that skilled attendance at delivery should be ensured. The call emphasised that all women should have access to contraception to avoid unintended pregnancies, all pregnant women should prepare and have access to skilled care at the time of birth, and all those with complications to have timely access to quality emergency obstetric care; backed up by transport in case of emergency referral [18]. The SMI hinges on six pillars—family planning, antenatal care (including birth preparedness and complication readiness), obstetric care (clean and safe delivery), Postnatal care, post-abortion care and sexually transmitted infection (STI) control; and supported by communication for behaviour change, primary health care, and equity for women [18]. Effective implementation of SMI components, if not constrained by social and cultural norms, beliefs, and attitudes, can translate to good maternity care and safe motherhood.

Interestingly, community participation (including men) is a veritable strategy for addressing socio-cultural factors affecting maternal health and improving access to healthcare in rural communities [19, 20]. However, Male involvement in maternity care and safe childbirth is complex and shaped by many socio-cultural factors, including availability, cultural beliefs and traditions [3, 5]. Engaging community members in implementing safe motherhood intervention can enhance SMI support systems through better interpersonal communication to dispel misconceptions and myths and bring about positive behavioural change. It can also enable equity for women by gaining the support and understanding of men. Furthermore, community participation aligns with the WHO task shifting policy focus for addressing the dearth of human resources for health, which is common in rural areas, in addition to helping take health care to the people [21]. Evidence has shown that community-based interventions that provide education to women, men and the community at large have the potential to increase demand for the utilisation of skilled care and improve safe motherhood practices [21]. Such improvement is envisaged to reduce delays in seeking care and thereby help reduce maternal mortality. This study assessed the effect of participatory-action research on male involvement as supportive partners to improve maternity care and safe motherhood in rural southeast Nigeria.

## Methods

### Study setting

The study was carried out in rural communities in Ebonyi state, southeast Nigeria. The state has thirteen local government areas (LGAs) ten of which are rural LGAs. According to the 2018 demographic survey, the state has the worst maternal health indices:—the least utilisation of ANC (70.3%), vaccination against neonatal tetanus (87.3%), delivery service from a skilled provider (58.3%), facility delivery (56.6%), postnatal services (50.2%); compared to other southeast states where the prevalence of 97.4%, 96.9%,94.9%,94.5% and 82.1% respectively were reported [8]. Most people of the State (over 75%) reside in rural areas, are poor, and lack formal education. These factors could contribute to the poor indices as urban, high wealth index, and educated women are more likely to receive maternal care from skilled providers [8] The inhabitants are mostly farmers and belong to trans-generational associations which serve as avenues for socialisation.

### Study design

A pre-post-intervention study was conducted in three phases: pre-intervention, intervention, and post-intervention. The first phase was a baseline assessment using a quantitative research method, the second phase was community-participatory intervention on safe motherhood, and the third was a post-intervention assessment as applied in the baseline.

### Research participants

The population included in the study comprised male partners of pregnant women, who are adults (eighteen years and above) and are permanent residents (who lived a minimum of three years) of the selected rural LGA and communities. Those excluded from the study were participants who declined consent or were unfit to participate due to severe medical conditions.

### Sample size calculation and sampling method

The sample size was calculated using the formula for comparing two proportions with a standard error of 5%, power of 80% and net intervention effect of 20.2% (proportion of male partners that escort pregnant women to ANC) [22]. A sample size of 396 was obtained after adjusting for a 20% loss to attrition and a design effect of 1.5.

A rural LGA was purposively selected. The LGA is one of the LGAs with the poorest maternal indices–the highest maternal death. A two-staged modified cluster sampling was used to recruit participants for the study. A cluster was defined as a village governed by an elected or appointed traditional head. In the first stage, a simple random sampling technique of balloting was used to select two from the five autonomous communities in the LGA. In the second stage, ten clusters (villages)–five from each community were selected using a simple random sampling technique of balloting. Households having pregnant women from each cluster were mapped out and consecutively selected. The nearest public facility from the main entrance of each cluster was identified as the starting point. With equal allocation to each cluster, all eligible participants (partners to pregnant women) were recruited to participate in the study until the desired sample size was reached. The sampling and recruitment of the study participants are illustrated in Fig 1.

### Data collection

A structured questionnaire (5-point Likert scale type) adapted from a previous study [22] was used to collect data by ten trained research assistants (community members) comprising five

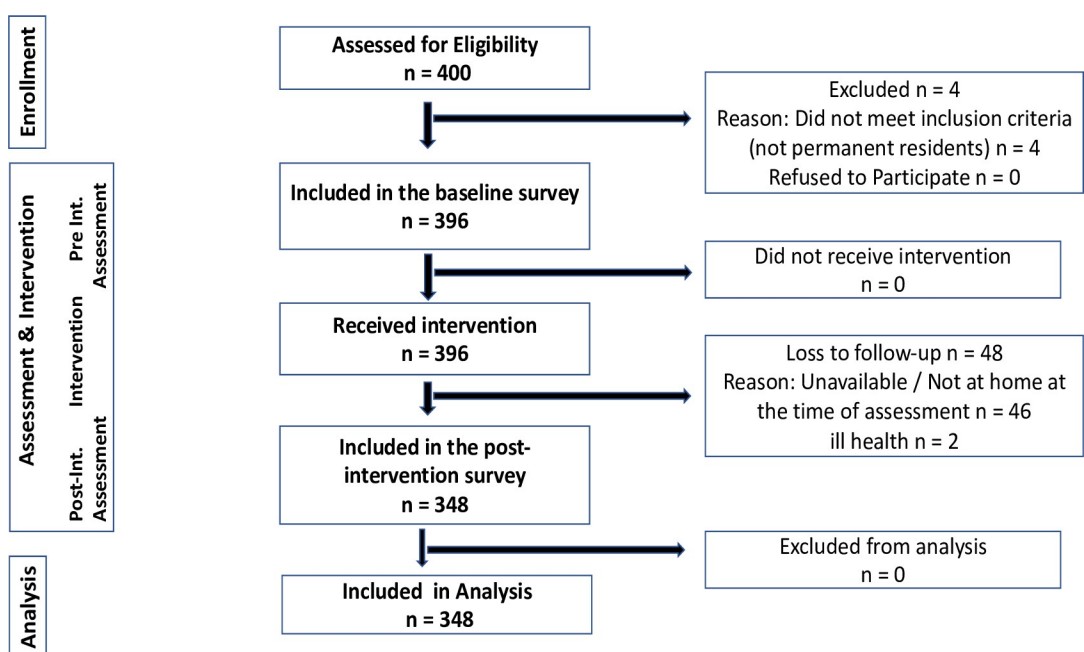

**Fig 1. Flow chart of the sampling and recruitment of the study participants.**

males and five females. The study instrument had four sections: -i) socio-demographic characteristics including age, marital status, the highest level of education, religion, and occupation, ii) awareness of safe motherhood/birth preparedness and complication readiness, iii) perception of men towards maternity care and safe motherhood comprising fourteen questions, and iv) male involvement/practices in maternity care and safe motherhood in terms of pregnancy care (comprising seven questions) and birth preparedness and complication readiness (comprising six questions). Awareness was determined as the proportion of respondents that answered "yes" to having heard about safe motherhood/birth preparedness and complication readiness. Responses to the questions on perceptions and practices were given corresponding Likert scale scores of one to five. This was summed up to get the mean scores/mean net rating (MNR), with the MNR of 3.0 as the logical neutral point [20, 23]. Good perception and good practice towards maternity care and safe motherhood were determined by MNR >3.0 To ensure validity, the questionnaire was pretested among 20 males in another community not selected for the study. The questionnaires were administered to the participants at their homes for convenience after getting informed written consent, before the intervention and six months post-intervention.

## Intervention

A community-participatory intervention was implemented comprising advocacy, and training of community volunteers, who then educated male partners of pregnant women on safe motherhood and facilitated emergency saving and transport schemes.

Advocacy visits were made to community leaders, traditional rulers and cabinet members, village heads and community association leaders to create awareness and secure support for the programme. This was followed by a one-day training of twenty community volunteers, ten males and ten females by the researchers on the components and processes of the intervention. Thereafter, the trained community volunteers educated the study participants on the

misconceptions about pregnancy, maternity care, and safe motherhood, including birth preparedness and complication readiness practices. The training/education module was adapted from a previous study [22] and modified to suit the context. The topics covered include—i) myths and misconceptions about pregnancy, maternity care, and safe motherhood, ii) birth preparedness and complication readiness practices, and iii) the importance and processes of community participatory activities for safe motherhood. The education was provided to the study participants (and family members) through lectures, pictorials, and discussions, at their homes and at convenient times and was communicated in the local dialect. Each household education session lasted an average of 60 minutes. Furthermore, the community volunteers facilitated the implementation of the Emergency Fund Saving Scheme (EFSS) and Emergency Transport Schemes (ETS) to the leaders of community associations and road transport workers. These served as community activities to promote birth preparedness and complication readiness practices such as saving money and identifying transport. A detailed description of the intervention can be found in a previously published study [20].

## Ethical statement

Ethical approval was obtained from the Research Ethical Committee of Ebonyi State Ministry of Health (Ref No. SHOH/ERC/050/19). Permission was obtained from the traditional rulers and village heads of the communities. Informed written consent was obtained from participants by signing/thumbprint the consent form after a thorough explanation was given and understanding established. Participation was voluntary, and respondents were informed that they were at liberty to decline to participate or withdraw from the study with no consequences to them at any time. Confidentiality was assured to participants, and personally identifiable information like names was not captured.

## Data analysis

Statistical Package for Social Sciences (IBM-SPSS) for Microsoft Windows version 22 software was used for data entry and analysis. Descriptive statistics were presented as means and standard deviations for continuous variables, and frequencies and proportions for categorical variables. A comparison of the perceptions and practices towards maternity care and safe motherhood before and after the intervention was conducted to determine the mean difference using paired T-test. Statistical significance was set at p-value $< 0.05$.

## Definition of variables

Awareness was defined as the proportion of respondents that had heard about safe motherhood/birth preparedness and complication readiness.

Good perception towards maternity care and safe motherhood was determined by MNR $>3.0$, and poor perception by MNR $\leq 3.0$.

Good practice towards maternity care and safe motherhood was determined by MNR $>3.0$, and poor practice by MNR $\leq 3.0$.

## Results

This is a pre-post intervention study design that assessed the effect of participatory action research in improving male perceptions and practices towards maternity care and safe motherhood in rural communities. The study had 396 participants recruited at baseline and 348 at the post-intervention, giving a 12.1% attrition rate.

Table 1. Socio-demographic characteristics of men in rural Ebonyi communities.

| Variable | Frequency | Percent (%) |
|---|---|---|
| | (N = 396) | |
| **Age of respondents in years** | | |
| 20–29 | 96 | 24.3 |
| 30–39 | 153 | 38.6 |
| 40–49 | 120 | 30.3 |
| ≥50 | 27 | 6.8 |
| **Marital status** | | |
| Single | 9 | 2.3 |
| Married | 337 | 97.7 |
| **Highest educational level** | | |
| No formal education | 42 | 10.6 |
| Primary | 180 | 45.5 |
| Secondary | 162 | 40.9 |
| Tertiary | 12 | 3.0 |
| **Religion** | | |
| Christians | 342 | 86.4 |
| Traditional religion | 54 | 13.6 |
| **Occupation** | | |
| Farming | 162 | 40.9 |
| Artisan | 150 | 37.8 |
| Trading | 36 | 9.1 |
| Civil service | 24 | 6.1 |
| Unemployed | 24 | 6.1 |

The socio-demographic characteristics of the participants indicate that the highest proportion of the respondents, 38.6%, were in the age group 30–39 years. About 45% had primary education, while the least proportion, 3.0%, have attained tertiary education. Most of the respondents were married (97.7%) and Christians (86.4%) (Table 1).

Most participants (75.8%) were aware of safe motherhood. The main source of information was radio (45.0%) and health workers (31.8%) (Table 2).

The participants had low mean scores (less than 3.0) in more than one-third of the variables at baseline. The highest mean score of 4.51 (0.53) was recorded for women needing rest and special nutrition during pregnancy and the lowest mean score of 1.92 (0.83) for the necessity of male partners to accompany pregnant women to ANC visits.

After the intervention, there was a significant increase in the mean score of perceptions that women should plan ahead of time where to give birth (p = 0.008), how to get to the place she will give birth (p = 0.004), educating male partners on maternal health will enhance male involvement (p = 0.007), the necessity for male partners to accompany pregnant women to antenatal care visits, and that it is wrong for pregnant women to be beaten by male partners for any reason (p<0.001) (Table 3).

There was good practice, as shown by the sub-mean score of >3.0 for almost all the variables of maternal care and birth preparedness and complication readiness. Concerning maternal care, a significant increase was noted after the intervention in the mean scores of permitting the partner to attend ANC (p<0.001), helping the partner with household chores during pregnancy (p = 0.001), and discussing/encouraging the partner to have facility delivery (p<0.001). Birth preparedness and complication readiness components–saving money,

**Table 2. Awareness of safe motherhood and the main source of information among men in the rural Ebonyi community.**

| Variable | Frequency | Percent (%) |
|---|---|---|
| | (N = 396) | |
| **Awareness of safe motherhood** | | |
| Yes | 300 | 75.8 |
| No | 96 | 24.2 |
| **Source of information** | | |
| Radio | 163 | 41.2 |
| Health workers/outreach | 126 | 31.8 |
| Friends/Neighbours | 59 | 14.8 |
| Television | 24 | 6.1 |
| Print media | 24 | 6.1 |

identifying transport, blood donor, skilled provider, where to go for health care and preparing birth kits increased significantly after the intervention in all the discrete variables as well as the composite mean score (p<0.001) (Table 4).

## Discussion

The study was a pre-post intervention study that aimed to assess the effect of engaging men through community-participatory intervention to improve the perception and practices of maternity care and safe motherhood. The result showed that three-quarters of the study participants have heard about safe motherhood, birth preparedness and complication readiness. This finding resonates with a study conducted in rural Bangladesh which showed that two-thirds of partners to pregnant women were aware that women have special rights related to pregnancy and childbirth [12]. The high awareness may be related to the common and frequent exposure of the rural populace to health outreach programmes from health workers,

**Table 3. Perception toward maternity care and safe motherhood among men in rural communities pre- and post-intervention.**

| Perceptions towards maternity care | Pre- intervention | Post- intervention | Mean difference |
|---|---|---|---|
| | Mean (SD) N = 396 | Mean (SD) N = 348 | (p-value for paired T-test) |
| Women should plan ahead where they will give birth | 4.33 (0.63) | 4.53 (0.51) | 0.20 (0.008) * |
| A woman should plan-ahead how she will get to the place she will give birth. | 4.33 (0.61) | 4.54 (0.51) | 0.21 (0.004) * |
| Family planning is necessary for maternal health | 4.49 (0.54) | 4.55 (0.54) | 0.06 (0.394) |
| Woman needs rest/special nutrition during pregnancy | 4.51 (0.53) | 4.56 (0.49) | 0.05 (0.421) |
| Pregnancy/childbirth are women's matter that should not involve male partners | 4.16 (1.05) | 4.38 (0.88) | 0.22 (0.074) |
| Educating male partners in maternal health will enhance male involvement | 2.72 (1.30) | 3.12 (0.97) | 0.40 (0.007) * |
| Cultural belief restraints male partner from getting involved in maternal care | 2.35 (0.80) | 2.45 (1.11) | 0.09 (0.453) |
| Pregnant women and male partners should discuss issues concerning pregnancy regularly | 3.59 (1.12) | 3.78 (0.85) | 0.18 (0.152) |
| It's necessary for male partners to accompany pregnant women to ANC | 1.92 (0.83) | 2.67 (1.27) | 0.75 (<0.001) * |
| It is necessary for male partners to accompany pregnant women to give birth | 2.60 (1.34) | 2.72 (1.31) | 0.11 (0.491) |
| There should be joint decision (male partner and pregnant woman) in seeking care | 3.94 (0.94) | 4.07 (0.47) | 0.13 (0.182) |
| Male partners should help pregnant women with household chores during pregnancy | 4.22 (0.85) | 4.32 (0.66) | 0.10 (0.273) |
| It is wrong for a pregnant woman to be beaten by her male partner for any reason | 2.27 (1.12) | 2.90 (1.27) | 0.63 (<0.001) * |
| Health facility is the best place for skilled care during pregnancy and childbirth | 3.99 (0.36) | 4.03 (0.55) | 0.03 (0.514) |
| Overall perception mean score | 3.66(0.48) | 3.67 (0.37) | 0.003 (0.951) |

* Statistically significant; ANC—antenatal care.

**Table 4. Safe motherhood practices among men in rural Ebonyi communities at pre and post-intervention.**

| Variables | Pre intervention | Post intervention | Mean difference (p-value for paired T-test) |
|---|---|---|---|
| | Mean (SD) N = 396 | Mean (SD) N = 348 | |
| **The extent of maternal care** | | | |
| Permit your partner to attend antenatal care | 3.42 (1.35) | 3.60 (1.19) | 0.19 (0.251) |
| Accompany partner to ANC at least once during this pregnancy | 4.09 (1.09) | 4.81 (0.57) | 0.72 (<0.001) * |
| Help partner with household chores during pregnancy | 4.18 (1.04) | 4.58 (0.77) | 0.39 (0.001) * |
| Discuss pregnancy issue, childbirth and seeking care | 4.11 (1.04) | 4.30 (0.94) | 0.19 (0.123) |
| Allow for joint decision on maternal issues | 4.01 (1.11) | 4.22 (1.05) | 0.22 (0.117) |
| Encourage/support partner for family planning | 3.42 (1.35) | 3.62 (1.19) | 0.20 (0.151) |
| Discuss/encourage partner to have facility delivery | 4.11 (1.08) | 4.55 (0.82) | 0.45 (<0.001) * |
| Composite mean score | 3.98(0.76) | 4.34(0.58) | 0.36 (<0.001) * |
| **Adequacy of birth preparedness and complication readiness** | | | |
| Identify transport | 3.76 (1.52) | 4.46 (0.81) | 0.70 (<0.001) * |
| Save money | 4.19 (1.04) | 4.90 (3.76) | 0.71 (<0.001) * |
| Identify blood donor | 2.67 (1.64) | 4.01 (1.35) | 1.34 (<0.001) * |
| Identify skilled provider | 3.26 (1.47) | 4.02 (0.95) | 0.76 (<0.001) * |
| Identify where to go for healthcare | 3.66 (1.66) | 4.13 (1.03) | 0.47 (0.006) * |
| Prepare birth Kit | 4.33 (1.18) | 4.89 (0.39) | 0.56 (<0.001) * |
| Composite mean score | 3.68(0.99) | 4.47(0.82) | 0.79 (<0.001) * |

* Statistically significant; ANC—antenatal care.

which incidentally was a main source of information (in addition to radio) in this study. Furthermore, this study revealed that men accompany their partners to ANC visits. These interactions with the health system and possibly the education the men receive during the ANC visits (if any) may have contributed to the high awareness.

This study revealed before the intervention, that men have a good perception of some maternity care and safe motherhood components such as the need to prepare for birth, facility delivery, a joint decision in maternal health, special nutrition in pregnancy and helping with household chores. Corroborating our findings, a qualitative study conducted in Tanzania reported that preparation for birth was perceived as necessary to facilitate good health care and health facility delivery was viewed positively [24]. In rural Rwanda, it was revealed that participants perceived the importance of family assistance and attending ANC in facilitating birth preparedness and complication readiness and enhancing professional care at birth [25]. Nonetheless, our findings also, showed that men have poor perceptions that educating men in maternal health will enhance male involvement, the necessity for male partners to accompany their pregnant women to ANC visits and childbirth and that it is wrong for male partners to beat pregnant women for any reason. Furthermore, there was poor perception towards the sociocultural influences and restraints of male involvement in maternity care, contrary to earlier findings where factors such as custom, lack of perceived need, distance, lack of transport, lack of permission, and cost, were perceived as influencing maternal health [26–28]. Therefore, a targeted approach like community participatory intervention, which can impact these sociocultural factors, is of utmost importance.

After the community participatory safe motherhood intervention, there was an improvement in the perception towards preparing for birth, supporting facility delivery, a joint decision in maternal health, special nutrition in pregnancy, and helping with household chores. This corroborates the improvement reported in a pre-post intervention study in Benin City,

Nigeria, concerning an educational session's impact on men's perception regarding safe motherhood [29]. Our findings also align with earlier studies in sub-Saharan Africa which reported that participatory intervention was effective in improving obstetric perceptions [22, 30, 31]. Nevertheless, more effort is needed in modifying these sociocultural norms affecting males' perception towards maternity care. This is necessary because even though the intervention may have played a decisive role in positively changing some misconceptions, some perceptions, like not seeing anything wrong with a pregnant woman being beaten by the male partner for any reason, (although significantly improved), were still poor. This may be related to the study participants' perception towards the cultural influences and restraints of men in maternal health, which remained poor even after the intervention. Sharing the research findings, more engagement, re-orientation, and enlightenment of the communities concerning the relationship of negative perceptions, norms and values with obstetric complications/emergencies and its implications on maternal health is worthwhile.

This study revealed good safe motherhood practices such as male partners accompanying their partner to ANC and childbirth, helping with household chores, discussing pregnancy issues, and allowing for a joint decision on maternal issues, similar to an earlier study [15]. The importance of joint decision-making was highlighted in a study in South-Western Uganda which reported that women's decision-making on the location of birth in consultation with partners/relatives showed a significant effect on birth preparedness practices and choosing assistance by a skilled birth attendant at birth [15]. Furthermore, the study participants showed good practices of birth preparedness and complication readiness, consistent with the findings of a study in Uganda [15]. However, the good safe motherhood practices noted in this study are contrary to the results of a cross-sectional study that assessed male participation in maternity care in northern Nigerian communities, which found that less than one-third of the sampled men ever accompanied their spouses for maternity, saved for emergencies, prepared for transportation during labour, and less than a tenth prepared for skilled assistance during delivery [32]. The study, however, attributed the findings to the study participants' young paternal age and low educational status [32]. Unfortunately, the practice of identifying potential blood donors was poor, consistent with the result of a previous study [12]. This is worrisome as the vital role of blood in pregnancy cannot be overemphasised.

After the intervention, there was an improvement in maternity care practices, including accompanying partners to attend ANC, facility delivery, and helping partners with household chores during pregnancy. Consistent with our finding, a quasi-experimental study conducted in Benin, where men participated in a joint educational session on safe motherhood, showed that the mean composite scores increased significantly [29]. Our results also resonate with a study on the impact of an intervention on maternal health outcomes in northern Nigerian which showed that women with standing permission from their husbands to go to the health centre and attend antenatal care doubled and delivery with a skilled birth attendant and health care utilisation increased in the intervention communities [33]. A similar positive finding was shown in a study in rural Bangladesh which reported that approximately three-quarters of male partners discussed maternal health with their pregnant partners, accompanied them to antenatal and post-natal care, and were present at the birthplace during birth [12]. Consistent with our findings, a community-based educational intervention implemented by trained community health workers in rural Tanzania showed increased male involvement in maternal health with marked improvement in the proportion of men accompanying their wives to antenatal and delivery and shared decision-making for the place of delivery [34]. Male partners accompanying their pregnant partners were found to be positively associated with women receiving ANC from a medically trained provider, giving birth at a health facility, receiving PNC and seeking care from a medically trained provider for obstetric complications [12].

Therefore, males accompanying pregnant partners to clinics should be advocated for inclusion into maternal health policy.

Furthermore, after the intervention, there was a significant improvement in birth preparedness and complication readiness practices such as saving money, identifying transport, blood donor, skilled provider, where to go for health care and preparing birth kits. Contrary to our finding, a quasi-experimental study on the effect of behavioural intervention on male involvement in birth preparedness in Northern Nigeria did not show a statistically significant increase in the practice of birth preparedness [10]. The qualitative data analysis revealed that their religious beliefs were not in favour of the practice of birth preparedness. So, the intervention did not increase male involvement in birth preparedness, likely due to religious misconceptions [10]. However, a study in southern Nigeria reported an increase in individual savings, group savings, and identification of a transporter following the community-based intervention [29]. A similar participatory action research to improve birth preparedness and health facility access in rural Uganda showed improved birth preparedness and more male involvement in maternal health [31, 35]. Saving groups and other saving modalities were strengthened, with money saved to meet transport costs, purchase other items needed for birth and other routine household needs. Also, linkages between savings groups and transport providers improved women's access to health facilities at a reduced cost [35]. Going by the improved practice to save money for health care shown among men in this study, there is a need to consolidate community insurance schemes which could help reduce primary healthcare costs and the disparity in health access [11].

Our finding showed that the community-participatory intervention improved males' perception and practices towards maternal health and safe motherhood. Earlier evidence reveals that community participation is an effective strategy in influencing individuals' and community perceptions, attitudes, and behaviour towards maternal health in rural areas [36]. It could be implied that community-participatory intervention is feasible and effective in improving male involvement in maternal health. It plays a decisive role in modifying the sociocultural norms affecting men's perceptions and practices towards pregnancy. There is a possibility that since the intervention implementers were the community members, communication may have been easier and better trust established, hence the positive effect. Studies have shown that interventions that employed a community-participatory approach were well accepted by the community because of good interpersonal communication, which influenced them to make health-promoting decisions and seek skilled health services [22, 30, 37]. In addition, using community volunteers as a task-shifting approach is worthwhile as it can enable rational redistribution of tasks among existing health workforce cadres, from highly qualified health workers to health workers with shorter training and fewer qualifications. This is especially important in rural areas where shortage and maldistribution of human resources for health is common [21]. This measure will enable more efficient use of the available health workers and improve personnel deficit, especially in the hard-to-reach and underserved areas. Important lessons drawn from these findings include integrating local context, men, and the wider community with the existing health systems.

The benefits of male involvement in maternal health cannot be over-emphasized. A systematic review reported that male involvement was associated with improved maternal health outcomes, such as reduced odds of postpartum depression, and improved utilisation of skilled maternal health services in developing countries [38]. A review of demographic health survey in India revealed that men's awareness about pregnancy-related care and a positive gender attitude enhances maternal health care utilisation and women's decision-making about their health care, and their presence during antenatal care visits markedly increases the chances of women's delivery in institutions [39]. Men's supportive stance is essential for improving women's world and enhancing maternal health, especially in a male-driven society.

## Limitations

The limitation of this study is that the practices were self-reported. As there is a strong understanding that some practices like facility delivery and birth preparedness are favourable behaviours, participants may want to provide socially desirable responses leading to bias. This was overcome by using community members as data collectors to establish better trust, extensively explaining the benefit of the research, and assuring participants of confidentiality.

## Conclusions

The results show variable perceptions towards maternity care and safe motherhood among men, which improved significantly after the intervention. There is good male involvement with maternity care, birth preparedness, and complication readiness, which improved further with the intervention. This interest and willingness should be leveraged, and males accompanying their pregnant partners to clinics should be advocated for inclusion into maternal health policy. Given the positive contributions of community members to maternity care, the government should integrate community health influencers and promoters into the healthcare systems to provide health services and reduce problems of personnel deficit and work overload, especially, in the hard-to-reach and underserved areas. Community linkages need to be strengthened through increased stakeholder engagement to enhance service delivery using community-oriented resource persons.

### Key messages

**1. Implication for policy makers.**

- To emphasise the importance of task shifting using community-oriented resource persons as a strategic mechanism for addressing health service access gaps, the deficit of skilled manpower for health and work overload, especially in rural and hard-to-reach areas.

- To highlight the possibility of synergy between researchers, policy makers and research users in generating context-specific evidence which is important for evidence-based policymaking.

- To emphasize that communities can be empowered with relevant health information and services to improve health using collaborative and participatory approaches thereby strengthening local community efforts.

**2. Implication for public.** The research revealed good male involvement in maternal health, which improved with user participation in the implementation. It showed improvement in males' perception towards maternity care and safe motherhood in relation to the importance of rest and healthy nutrition during pregnancy and birth preparedness. It reflected good maternity care practices in terms of joint decision-making, helping with household chores, accompanying spouses for antenatal care and facility childbirth which should be leveraged to improve maternal health. The research revealed adequate birth preparedness and complication readiness practices which improved post-intervention, such as saving money and identifying transport, skilled provider, health facility, and blood donors for emergencies, which should be sustained. The study highlights that community participatory strategy can enhance male involvement in maternity care and safe motherhood and should be further explored.

## Supporting information

**S1 Dataset. Irene Eze total data set.**
(SAV)

## Acknowledgments

The authors thank the community heads for their cooperation and appreciate the participants for partaking in the study.

## Author Contributions

**Conceptualization:** Irene Ifeyinwa Eze, Chigozie Jesse Uneke.

**Data curation:** Irene Ifeyinwa Eze, Edmund Ndudi Ossai, Lawrence Ulu Ogbonnaya, Chigozie Jesse Uneke.

**Formal analysis:** Edmund Ndudi Ossai.

**Methodology:** Irene Ifeyinwa Eze, Edmund Ndudi Ossai, Ifeyinwa Chizoba Akamike, Ijeoma Nkem Okedo-Alex, Lawrence Ulu Ogbonnaya.

**Project administration:** Ifeyinwa Chizoba Akamike, Ijeoma Nkem Okedo-Alex, Lawrence Ulu Ogbonnaya.

**Supervision:** Chigozie Jesse Uneke.

**Validation:** Irene Ifeyinwa Eze, Edmund Ndudi Ossai, Ifeyinwa Chizoba Akamike, Ijeoma Nkem Okedo-Alex, Chigozie Jesse Uneke.

**Visualization:** Irene Ifeyinwa Eze, Edmund Ndudi Ossai, Ifeyinwa Chizoba Akamike, Ijeoma Nkem Okedo-Alex, Lawrence Ulu Ogbonnaya.

**Writing – original draft:** Irene Ifeyinwa Eze.

**Writing – review & editing:** Edmund Ndudi Ossai, Ifeyinwa Chizoba Akamike, Ijeoma Nkem Okedo-Alex, Lawrence Ulu Ogbonnaya, Chigozie Jesse Uneke.

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
