## [Decision Letter · Decision Letter 0]

15 Aug 2022

PONE-D-22-01826Males’ perceptions and practices towards maternity care in rural southeast Nigeria: policy implication of participatory action research for safe motherhoodPLOS ONE

Dear Dr. Eze,

Thank you for submitting your manuscript to PLOS ONE. After careful consideration, we feel that it has merit but does not fully meet PLOS ONE’s publication criteria as it currently stands. Therefore, we invite you to submit a revised version of the manuscript that addresses the points raised during the review process. 

We look forward to receiving your revised manuscript.

Kind regards,

Comfort Z Olorunsaiye, Ph.D.

Academic Editor

PLOS ONE

Journal Requirements:

Reviewers' comments:

Reviewer's Responses to Questions

**Comments to the Author**

1. Is the manuscript technically sound, and do the data support the conclusions?

Reviewer #1: Yes

Reviewer #2: Yes

2. Has the statistical analysis been performed appropriately and rigorously? 

Reviewer #1: Yes

Reviewer #2: Yes

3. Have the authors made all data underlying the findings in their manuscript fully available?

Reviewer #1: Yes

Reviewer #2: Yes

4. Is the manuscript presented in an intelligible fashion and written in standard English?

Reviewer #1: Yes

Reviewer #2: Yes

5. Review Comments to the Author

Reviewer #1: This manuscript describes an interventional study that utilizes multiple components of a community-participatory program to improve male partner involvement in maternal and newborn care. Given that it is established that male involvement is positively associated with improved reproductive health outcomes in Sub-Saharan Africa, high quality evidence-based interventions to inform practice and policy are needed. I submit the following comments:

Major comments:

Introduction and throughout the manuscript: Suggest replacing "husbands" with male partners. The use of "husbands/wives" connotes legal marriage, which may not necessary be the case for most individuals given the setting under consideration. Male partners may be more inclusive language.

Methods

Suggest including a flow chart depicting sampling and recruitment activities to make it easier for readers to understand.

Line 119 - 123: Suggest including numbers to describe the poor maternal health indices in Ebonyi state to give a general audience more contextual information. Please provide the source of these metrics.

Line 152: "Study instrument": It would be helpful if the authors included a copy of the survey instrument administered for data collection. If not feasible, please provide a summary of how the key variables were defined. For instance, how was awareness of safe motherhood assessed?

Given that the survey instrument was adapted from a previous study, did the authors assess the validity and reliability of the tool for this population? What was the Cronbach's alpha for this population?

Line 160 - 173: In describing the intervention, the authors reported information from a previous published study that was focused on improving birth preparedness and complication readiness through a behavioral change intervention. It would be helpful for readers to briefly explain and focus on the specific components of this intervention that are targeted at male partners to improve their perceptions and attitudes toward safe motherhood.

Was the goal to "train" the study participants or educate them? Please clarify.

Also, how many of the male partners received the intervention? How long were the educational sessions for? Please clarify. This would help readers determine reach and fidelity of the intervention.

There is no mention of the mean difference in the analytical section. How was it computed and for what purpose?

Results

Suggest summarizing the findings as narratives rather than using the variable names to report the results.

Discussion

In introducing the premise of the study, the authors alluded to the low involvement of men in maternal health; notably their absence during ANC visits. However, in this study, the authors found that a high proportion of men were engaged with the healthcare system from accompanying their female partners during ANC visits. This may have led to the high level of awareness of maternity care and safe motherhood among the respondents. It is also likely that the education the men received (if any) during the ANC visits may have improved some of the other domains. This should be discussed.

Line 271 - 273:...high mean scores at baseline for maternity care variables like...: In the discussion section, the focus should be on contextualizing these findings and not be repetitive of the results section.

Similar comments for Lines 287 and 304

Minor comments

I would suggest the authors review the manuscript critically for grammar and clarity.

Reviewer #2: There is need to proof read and correct some grammatical errors in the whole article ( for example lines 35, 76, correct Sub- Sahara Africa to Sub- Sub-Saharan Africa in the whole work, replace bracket in line 137 and 138 with comma)

Abstract

Line 47: Provide justification for the recommendation that community health influencers/ promoters should be integrated into healthcare system. Did your study show that they helped to reduce personnel deficit problems?

Introduction

Line 61: Provide a reference for Nigeria contributing 14% to global maternal mortality; not NDHS

Line 62: Reference the WHO document where the statement was gotten from not NDHS

Lines 70-72: Kindly rephrase for better context

Lines 76- 82: Cite the papers that were referenced in the work of Rahman et. al.

Lines 90- 92: The studies cited are community based studies, what is the evidence that facility- based dissemination of messages is the reason for the poor awareness of safe motherhood in Nigeria? How did you arrive at the conclusion that facility based dissemination of messages has been considered as the only reliable means of conveying safe motherhood information in Nigeria?

The concepts of maternity care, safe motherhood, community participation have not been explained in the introduction. Additionally, how do perception and practice of maternity care impact on safe motherhood especially in Rural Areas and how do all these including male involvement relate to participatory action research? What is the relationship between maternal care, birth preparedness, complication readiness; and safe motherhood? These need to be clearly stated in the introduction.

Methods

line 140: What were the maternal indices used in the selection and how did you arrive at the rating of ''poorest'? Why was a probability sampling technique not used in selecting the LGAs?

How were the effects of the three different interventions on safe motherhood measured?

Line 187: Association between variables, change this to mean difference between ......

Results

Occupation sums up to 100.1, kindly correct this

Delete ‘’Table XX shows’’ from the beginning of the sentence and substitute it with (Table XX) at the end of the sentence

Discussion

Line 240: How was high awareness obtained? What were the cut-offs for high or low awareness? It’s not stated in the methodology.

There is a need to revise the discussion to include possible explanations on the study findings, public health implications of the findings and not merely reporting study findings.

Also, it is worthwhile to discuss why important variables such as perception of males on pregnant wife been beaten by her husband for any reason still remained poor despite interventions.

The discussion focused mostly on studies whose findings were similar to those of the authors, there’s a need to also include studies with contrary findings.

Limitations

How was the first limitation overcome? The second limitation should be deleted.

Conclusion

Line 368: How did you arrive at the conclusion that community members have positive contribution to maternal health? What aspect of your study suggests this? Community volunteers were trained to deliver the training on safe motherhood to men, how did you measure the impact of these volunteers on male involvement? How do we know that the positive change observed post intervention was not due to any of the other two interventions?

6. PLOS authors have the option to publish the peer review history of their article (what does this mean?). If published, this will include your full peer review and any attached files.

Reviewer #1: No

Reviewer #2: No

---

## [Author Response · Author response to Decision Letter 0]

8 Jan 2023

Reviewer #1: 

Major comments:

Introduction and throughout the manuscript: Suggest replacing "husbands" with male partners. The use of "husbands/wives" connotes legal marriage, which may not necessary be the case for most individuals given the setting under consideration. Male partners may be more inclusive language.

Thank you for the suggestion. The word “husbands” has been replaced with “male partners” throughout the manuscript.

Methods

Suggest including a flow chart depicting sampling and recruitment activities to make it easier for readers to understand.

The suggestion is well taken. A flow chart depicting sampling and recruitment activities has been included, on page 8

Line 119 - 123: Suggest including numbers to describe the poor maternal health indices in Ebonyi state to give a general audience more contextual information. Please provide the source of these metrics.

The proportions to describe the poor maternal health indices in Ebonyi State and the source has been included; see lines 143 – 152. Thank you. 

Line 152: "Study instrument": It would be helpful if the authors included a copy of the survey instrument administered for data collection. If not feasible, please provide a summary of how the key variables were defined. For instance, how was awareness of safe motherhood assessed? 

A summary of how the variables were defined has been indicated in lines 184-196, and 242 – 247. Thank you

Given that the survey instrument was adapted from a previous study, did the authors assess the validity and reliability of the tool for this population? What was the Cronbach's alpha for this population?

The survey instrument was not wholly adopted, but rather adapted from previous studies. Hence, the validity and reliability of the tools from previous studies were not assessed. However, this study's instrument was validated by pretesting in another community and modified to suit the context, 196 - 197.

Line 160 - 173: In describing the intervention, the authors reported information from a previous published study that was focused on improving birth preparedness and complication readiness through a behavioral change intervention. It would be helpful for readers to briefly explain and focus on the specific components of this intervention that are targeted at male partners to improve their perceptions and attitudes toward safe motherhood. 

Your view is well taken. Specific components of the intervention targeted at male partners to improve their perceptions and practices toward safe motherhood, which include advocacy for community participation, health education on pregnancy myths, maternity care and safe motherhood, facilitation of emergency transport and health savings schemes, have been included. See lines 202 – 223. Full details of the intervention have been published as referenced [21]

Was the goal to "train" the study participants or educate them? Please clarify.

The goal was to educate the study participant. However, community members/volunteers were trained to conduct the educational sessions and facilitate the community activities that promote safe motherhood for the study participants, as a community participatory process, lines 209 - 218

Also, how many of the male partners received the intervention? How long were the educational sessions for? Please clarify. This would help readers determine reach and fidelity of the intervention.

Three hundred and ninety-six male partners (and family members) received the intervention. Each household educational session lasted for an average of 60 minutes. Full details of the intervention have been published as referenced [21].

There is no mention of the mean difference in the analytical section. 

Thank you for drawing my attention to the omission. The mean difference has been added to the analysis section (lines 239 - 240). 

How was it computed and for what purpose?

The mean difference was computed by comparing the mean scores of the variables at the pre-intervention stage with the mean scores at the post-intervention stage using paired T-test. The purpose was to determine the effect of the intervention; if there is a statistically significant difference in the perceptions and practices following the intervention among the study participants

Results

Suggest summarizing the findings as narratives rather than using the variable names to report the results.

Thank you for the suggestion. The findings have been summarised as narratives

Discussion

In introducing the premise of the study, the authors alluded to the low involvement of men in maternal health, notably their absence during ANC visits. However, in this study, the authors found that a high proportion of men were engaged with the healthcare system from accompanying their female partners during ANC visits. This may have led to the high level of awareness of maternity care and safe motherhood among the respondents. It is also likely that the education the men received (if any) during the ANC visits may have improved some of the other domains. This should be discussed.

 Thank you for the input. Your suggestions have been reflected. Lines 301 -306

Line 271 - 273:...high mean scores at baseline for maternity care variables like...: In the discussion section, the focus should be on contextualizing these findings and not be repetitive of the results section. 

 Thank you. The sessions under the discussions have been modified, lines 341-343

Similar comments for Lines 287 and 304

Thank you. The sessions have been modified; lines 358 and 379

Minor comments

I would suggest the authors review the manuscript critically for grammar and clarity.

The comment is well noted and appreciated. The manuscript has been reviewed critically for grammar and clarity and corrected accordingly. Thank you 

Reviewer #2: 

There is need to proof read and correct some grammatical errors in the whole article ( for example lines 35, 76, correct Sub- Sahara Africa to Sub- Sub-Saharan Africa in the whole work, replace bracket in line 137 and 138 with comma)

The comment is noted with thanks. The article has been proofread and the grammatical errors corrected

Abstract

Line 47: Provide justification for the recommendation that community health influencers/ promoters should be integrated into healthcare system. Did your study show that they helped to reduce personnel deficit problems?

The study used community volunteers as resource persons to implement the intervention – providing health education and facilitating community activities (emergency fund savings and emergency transport), as elements of birth preparedness and complication readiness. As the results showed that community volunteers providing these services brought about improvement in males’ perception, maternity care and safe motherhood practices., it is suggested that integrating the community volunteers into the health systems as community-oriented resource persons can help reduce health personnel deficit and workload, especially in the rural areas where skilled workers are usually of short supply. 

Introduction

Line 61: Provide a reference for Nigeria contributing 14% to global maternal mortality; not NDHS

Reference added [1]; thank you

Line 62: Reference the WHO document where the statement was gotten from not NDHS

WHO document referenced added [2]; thank you 

Lines 70-72: Kindly rephrase for better context

The section has been rephrased, lines 74 – 77. Thank you

Lines 76- 82: Cite the papers that were referenced in the work of Rahman et. al.

Thank you. Appropriate reference cited [14 - Persplastiner M], Line 85

Lines 90- 92: The studies cited are community based studies, what is the evidence that facility- based dissemination of messages is the reason for the poor awareness of safe motherhood in Nigeria? How did you arrive at the conclusion that facility based dissemination of messages has been considered as the only reliable means of conveying safe motherhood information in Nigeria?

Thank you for the observation. The intention was not to portray factify-based dissemination of messages as the reason for the poor awareness nor as the only reliable means of conveying health information, rather, it is as supposedly the commonest source for accessing health information and services. The session has been corrected/modified for clarity. Lines 97 - 99

The concepts of maternity care, safe motherhood, community participation have not been explained in the introduction. Additionally, how do perception and practice of maternity care impact on safe motherhood especially in Rural Areas and how do all these including male involvement relate to participatory action research? What is the relationship between maternal care, birth preparedness, complication readiness; and safe motherhood? These need to be clearly stated in the introduction.

Thank you for the observation. The concept of maternity care, birth preparedness, complication readiness, safe motherhood, and community participation has added to the introduction. 

Positive perceptions and good practices towards maternity care can translate to safe motherhood, especially in rural areas where misperceptions, myths about pregnancy and poor value for women’s rights in a men-dominated patriarchal system are rife. 

Community participation is a veritable strategy for addressing socio-cultural factors affecting maternal health. The participatory action research relates to the engagement of the community members – men, women, and community associations, in the programme (both in data collection and implementation of the intervention). Community volunteers were trained as research assistants to collect the data, provide health education aimed at dispelling misperceptions and myths about pregnancy to the study participants, and facilitate community activities (emergency savings and transport schemes) that promote safe motherhood

Methods

line 140: What were the maternal indices used in the selection and how did you arrive at the rating of ''poorest'? Why was a probability sampling technique not used in selecting the LGAs?

How were the effects of the three different interventions on safe motherhood measured?

The maternal indices used in the study area selection is maternal death, line 169. The selected LGA has the highest proportion of maternal deaths in the state. A probability sampling technique was not used in selecting the LGAs because of the need to ensure that the intervention is implemented at the place it is most needed.

The intervention was assessed as one composite safe motherhood intervention comprising- advocacy, health education and facilitation of community activities to enhance safe motherhood, not as three separate interventions

Line 187: Association between variables, change this to mean difference between ......

Thank you for the input, association between variables has been changed to mean difference, lines 238 – 24

Results

Occupation sums up to 100.1, kindly correct this

Corrected, to sum up to 100. Thank you

Delete ‘’Table XX shows’’ from the beginning of the sentence and substitute it with (Table XX) at the end of the sentence

Revised so that the table number is cited at the end of the sentence. Thank you

Discussion

Line 240: How was high awareness obtained? What were the cut-offs for high or low awareness? It’s not stated in the methodology.

Thank you for the observation. Measurement of awareness has been added to the methodology. Awareness is defined as the proportion of respondent that has heard about the concept of safe motherhood/birth preparedness and complication readiness, lines 242-243.

There is a need to revise the discussion to include possible explanations on the study findings, public health implications of the findings and not merely reporting study findings.

This comment is well taken. The discussion section has been revised accordingly with the inclusion of the public health implications of the findings in relevant ares. 

Also, it is worthwhile to discuss why important variables such as perception of males on pregnant wife been beaten by her husband for any reason still remained poor despite interventions.

Thank you for the input. This discussion has been included, in lines 331-340

The discussion focused mostly on studies whose findings were similar to those of the authors, there’s a need to also include studies with contrary findings.

The comment is well taken. However, five studies that had findings at variances with those of the authors were cited –[27,28,29] (on perceptions, lines 318-321), and [33], [10] (on the practice of maternity care, birth preparedness and complication readiness; lines 349-555, 381-384)

Limitations

How was the first limitation overcome? The second limitation should be deleted.

Thank you. The limitation has been revised. It was overcome by establishing trust, explaining the research's benefit, and assuring confidentiality by using community members as data collectors 

Conclusion

Line 368: How did you arrive at the conclusion that community members have positive contribution to maternal health? What aspect of your study suggests this? Community volunteers were trained to deliver the training on safe motherhood to men, how did you measure the impact of these volunteers on male involvement? How do we know that the positive change observed post intervention was not due to any of the other two interventions?

The intervention was one composite intervention – a safe motherhood intervention implemented using multiple components. I viewed that the community members had a positive contribution to maternity care because of their full involvement in the intervention implementation - advocacy, safe motherhood education and community activities which brought about improvement in the perception and practices towards safe motherhood. Thank you

---

## [Editor Report · Decision Letter 1]

16 Feb 2023

Males’ perceptions and practices towards maternity care in rural southeast Nigeria: policy implication of participatory action research for safe motherhood

PONE-D-22-01826R1

Dear Dr. Eze,

We’re pleased to inform you that your manuscript has been judged scientifically suitable for publication and will be formally accepted for publication once it meets all outstanding technical requirements.

Kind regards,

Comfort Z Olorunsaiye, Ph.D

Academic Editor

PLOS ONE
---

## [Editor Report · Acceptance letter]

20 Mar 2023

PONE-D-22-01826R1 

Males’ perceptions and practices towards maternity care in rural southeast Nigeria: policy implication of participatory action research for safe motherhood 

Dear Dr. Eze:

I'm pleased to inform you that your manuscript has been deemed suitable for publication in PLOS ONE. Congratulations! Your manuscript is now with our production department. 

Kind regards, 

on behalf of

Dr. Comfort Z Olorunsaiye 

Academic Editor

PLOS ONE